# Progression of Diabetic Kidney Disease and Gastrointestinal Symptoms in Patients with Type I Diabetes

**DOI:** 10.3390/biomedicines11102679

**Published:** 2023-09-29

**Authors:** Aleksejs Fedulovs, Lilian Tzivian, Polina Zalizko, Santa Ivanova, Renāte Bumane, Jana Janeviča, Lelde Krūzmane, Eduards Krustins, Jelizaveta Sokolovska

**Affiliations:** 1Faculty of Medicine, University of Latvia, Jelgavas Street 3, LV 1004 Riga, Latvia; aleksejs.fedulovs@lu.lv (A.F.); liliana.civjane@lu.lv (L.T.); polinazalizko@inbox.lv (P.Z.); ivanova63@inbox.lv (S.I.); renate.bumane@gmail.com (R.B.); janevica.jana@gmail.com (J.J.); lelde.kruzmane@gmail.com (L.K.); 2Pauls Stradins Clinical University Hospital, Pilsoņu Street 13, LV 1002 Riga, Latvia; eduards.krustins@gmail.com

**Keywords:** type 1 diabetes, diabetic kidney disease, gastrointestinal symptoms, calprotectin, gastrointestinal endoscopy

## Abstract

(1) Background: Little research is conducted on the link between diabetic kidney disease (DKD) progression and diabetic gastroenteropathy in type 1 diabetes (T1D). (2) Methods. We performed a cross-sectional study with 100 T1D patients; 27 of them had progressive DKD, defined as an estimated glomerular filtration rate (eGFR) decline ≥3 mL/min/year or increased albuminuria stage, over a mean follow-up time of 5.89 ± 1.73 years. A newly developed score with 17 questions on gastrointestinal (GI) symptoms was used. Faecal calprotectin was measured by ELISA. Lower GI endoscopies were performed in 21 patients. (3) Results: The gastrointestinal symptom score demonstrated high reliability (Cronbach’s α = 0.78). Patients with progressive DKD had higher GI symptom scores compared to those with stable DKD (*p* = 0.019). The former group demonstrated more frequent bowel movement disorders (*p* < 0.01). The scores correlated negatively with eGFR (r = −0.335; *p* = 0.001), positively with albuminuria (r = 0.245; *p* = 0.015), Hba1c (r = 0.305; *p* = 0.002), and diabetes duration (r = 0.251; *p* = 0.012). Faecal calprotectin levels did not differ between DKD groups significantly. The most commonly reported histopathological findings of enteric mucosa were infiltration with eosinophils, lymphocytes, plasmacytes, the presence of lymphoid follicles, and lymphoid aggregates. Conclusion: The progression of DKD is positively correlated with gastrointestinal symptoms; however, more research is needed to clarify the causal relationships of the gut-kidney axis in T1D.

## 1. Introduction

The prevalence of type 1 diabetes (T1D) is constantly rising throughout the world. T1D is associated with increased morbidity and mortality, mainly due to its vascular complications, including diabetic kidney disease (DKD), which is the main reason for end-stage renal disease (ESRD) in the developed world [1]. Despite the progress in the understanding of the pathophysiology of DKD [2]., and constantly improving diabetes care to control for the main clinical risk factors of the disease, it is still largely unclear which patients are at increased risk of rapidly progressing DKD (estimated glomerular filtration (eGFR) loss of ≥3 mL/min/1.73^2^ per year [1].

Diabetic gastroenteropathy is a condition characterized by various oesophageal, gastric, intestinal, and anorectal symptoms caused by diabetes [3,4]. Gastrointestinal symptoms in T1D are twice as common as in the general population and were associated with worse glycaemic control and lower quality of life in studies [5,6]. A hypothesis was proposed that diabetic gastroenteropathy might predispose affected individuals to augmented intestinal inflammation and permeability and higher levels of systemic inflammation [7]. Low-grade systemic inflammation is in turn associated with the progression of complications of diabetes, including DKD [8,9]. To support the concept of this so-called gut-kidney axis, it was described that patients with T1D and macroalbuminuria have a higher level of the faecal inflammatory marker calprotectin compared to normoalbuminuric patients [10,11]. Unfortunately, the latter studies did not report symptoms of gastroenteropathy. Existing data on possible associations between DKD and gastrointestinal symptoms in T1D is currently limited to ESRD patients [12]. In the latter case, severe electrolyte imbalance, uraemia, oedema of the mucosa, medications, and generally very poor health might potentiate the symptoms. That is why data obtained in ESRD can hardly be extrapolated to patients with initial DKD stages. 

Gastrointestinal symptoms are insufficiently recognised and investigated, both in clinical practise and research. Specifically, endoscopic examination of the gastrointestinal tract might be avoided by T1D patients despite indications due to the increased risk of hypoglycemia and other complications [13]. Indirectly supporting the latter statement, we found only two papers reporting data on histological and immunohistochemical examination of intestinal mucosal biopsies of patients with T1D [12,14].

To summarize, diabetic gastroenteropathy as assessed by gastrointestinal symptom scores, instrumental examinations, and biomarkers (such as calprotectin) remains a little studied factor predisposing to or potentiating the progression of DKD. However, such data are of extreme importance for clinical practice, as they could promote the development of novel treatment and prevention options for DKD [15].

To fill this gap, the aim of this work was to analyse the prevalence of gastrointestinal symptoms, history of previous gastrointestinal conditions, data from endoscopic gastrointestinal investigations, levels of faecal calprotectin in patients with T1D, and different velocities of DKD progression in a cross-sectional study. 

## 2. Materials and Methods

### 2.1. Patients and Ethics

This study is a part of the longitudinal LatDiane study, initiated in 2013 (and participating in the international InterDiane consortium). LatDiane recruits adult patients with T1D diagnosed before the age of 40 with insulin treatment initiated within one year of diagnosis and C-peptide levels below 0.3 nmol/L. Patients with a history of chronic kidney disease apart from DKD are excluded from LatDiane. Follow-up visits and re-assessments of the status of complications of diabetes take place every three years. Currently, more than 400 patients have been recruited, with approximately 150 having attended one or more follow-up visits. Study protocol includes recording diabetes and other disease histories, basic clinical and anthropometric investigation, collection of blood samples for preparation of serum and plasma and DNA extraction, collection of urine, as well as assessment of dietary and physical activity habits, socioeconomic factors, and psychologic condition with the help of questionnaires [16,17,18,19,20,21,22,23,24]. The protocol of the general LatDiane and sub-study devoted to gut health described here were approved by the Latvian Central Ethics Committee and received permissions No 01-29.1/3 (dated 10 July 2013), Nr. A-17/19-10-17 (dated 17 October 2019), and Nr. 01-29.1/2226 (dated 30 April 2020).

Recruitment of this study participants, biobanking, and sample storage were performed in accordance with the procedures of the Genome Database of the Latvian population [25] and are described in detail in [22]. This study is in line with the 1964 Declaration of Helsinki and its later amendments. Written informed consent was obtained from all study participants prior to inclusion in this study.

Recruitment for this study took place between 15 January 2021 and 31 August 2022 in Latvia, Riga, Pilsoņu 13 str., building 10 (in the rooms of the Laboratory for Personalized Medicine of the University of Latvia). Information about this study was disseminated via the webpage and social media of the University of Latvia.

Inclusion criteria for this study were: T1D duration of at least 8 years and available data on progression of DKD (at least three yearly serum creatinine measurements and albuminuria measurements available between the baseline visit of the LatDiane study in 2013–2019 and this study) (Figure 1).

As for exclusion criteria, they were: pregnancy, subjects with a history of inflammatory bowel disease (Crohn disease or Ulcerative colitis), coeliac disease, acute intestinal infection within 2 months of the planned faecal collection, asymptomatic coeliac disease (detected via screening of serum transglutaminase IgA antibodies), clinical signs of acute inflammation, and fever.

On the study day, patients were investigated for the collection of anthropometric measures, questionnaires were filled out, blood samples were collected, and subjects received instructions and vials for the collection of the faecal sample.

### 2.2. Clinical Investigation and Monitoring of Diabetic Complications and Co-Morbidities

Clinical investigation included assessment of weight and height to calculate the body mass index (BMI, weight (kg)/height (m)^2^). We also measured blood pressure. Patients with systolic blood pressure ≥ 140 mmHg (18.7 kPa) or diastolic blood pressure ≥ 90 mmHg (12.0 kPa) or a history of antihypertensive drug usage were defined as having arterial hypertension.

Smoking was self-reported in the questionnaire; the “smokers” group referred to patients currently smoking at least one cigarette per day. 

Assessment of cardiovascular disease (CVD) and complications of diabetes, such as retinopathy, neuropathy, and DKD, was based on medical files. We defined CVD as a history of acute myocardial infarction, coronary bypass/percutaneous transluminal coronary angioplasty stroke, amputation, or peripheral vascular disease. 

The albumin-to-creatinine ratio in morning spot urine samples was used for the definition of albuminuria at each study visit during the follow-up. The estimated glomerular filtration rate (eGFR) was calculated according to the Chronic Kidney Disease Epidemiology Collaboration (CKD-EPI).

ESRD was defined as eGFR < 15 mL/min/1.73 m^2^, dialysis, or kidney transplantation.

Progressive DKD was defined as an eGFR decline exceeding 3 mL/min/1.73 m^2^/year [2] and/or increase in albuminuria stage over the follow-up period. At least three yearly serum creatinine measurements were used for the calculation of at least three eGFR values during the follow-up period. These data were used for eGFR slope plotting. The number of serum creatinine measurements ranged from three to fifteen during the follow-up for each patient.

### 2.3. Blood Samples and Faecal Collection

The blood samples were collected via venous puncture. Blood samples and morning spot urine samples were sent for assessment of clinical markers (blood count, clinical chemistry, and albuminuria) to a certified clinical lab.

Except for several patients who could collect the stool sample during the recruitment visit, stool samples were collected within two weeks after blood collection and assessment of gastrointestinal symptoms. Participants collected their faecal samples at home using sterile collection tubes without buffer (the collection date and time were marked). Within 24 h samples were delivered to the laboratory for calprotectin measurement in unfrozen samples. 

Faecal calprotectin was measured by the Alegria^®^ Calprotectin Elisa kit (Organotech Diagnostika GmbH, Budapest, Hungary, REF ORG280) in a certified clinical lab. 

### 2.4. Monitoring of Gastrointestinal Disease and Symptoms

The history of gastrointestinal disease was self-reported via an assisted study questionnaire in the majority of cases, and this information was supported by medical files. For reporting in this paper, we defined upper gastrointestinal disease as gastritis/duodenitis, peptic ulcer, gastroesophageal reflux disease, gastroparesis, diaphragmal hernia, and *Haelicobacter pylori* infection; lower gastrointestinal disease was defined as hemorrhoids, appendicitis, coeliac disease, irritable bowel syndrome, and lactose intolerance; and liver and pancreas disease were defined as histories of fatty liver, pancreatitis, viral hepatitis, gallstones, diverticulitis, and irritable bowel syndrome. 

A newly developed scale questionnaire for assessment of gastrointestinal symptoms was filled out during this study visit. It was elaborated on the basis of the diabetes and bowel symptom questionnaire [26], which previously demonstrated good test-retest reliability and concurrent validity (median kappa: 0.63 and 0.47, respectively) for the gastrointestinal items. Our newly developed scale questionnaire included 17 questions about pain, discomfort, impaired bowel movement such as diarrhoea, and constipation (Appendix A). 

### 2.5. Endoscopy in Patients with Indications and “Red Flag” Symptoms

Patients who had increased calprotectin levels and persistent symptoms of gastrointestinal disorders in the last 2 months were referred to a gastroenterologist for evaluation of indications for endoscopic examination (colonoscopy and/or endoscopy according to indications). As indications for endoscopy, the following criteria were used: unexplained weight loss, clinical suspicion of inflammatory bowel disease, unclear abdominal pain, unclear anaemia (especially iron deficiency anaemia), unclear diarrhoea, constipation or changing bowel movements, bowel movements with mucus admixture, family history of inflammatory bowel disease or colorectal cancer, history of polyps, elevated faecal calprotectin >50 µg/g, or as a screening colonoscopy for subjects over the age of 50 [27,28,29]. Out of 100 T1D patients, 47 were selected to undergo endoscopic examination, and 21 of them accepted the invitation and went through the procedure. Endoscopic biopsy specimens were examined by a histopathologist. *H. pylori* was identified by rapid urease testing (RUT) and histological staining (Giemsa). 

Patients who consented to lower endoscopy received recommendations from endocrinologists for preparation for the procedure based on international recommendations [13,30,31,32]. It included advice for food choices in the week preceding colonoscopy (“white foods”, avoidance of fibre-containing and red foods, etc.), frequent blood glucose monitoring during the reduced nutritional intake period, fasting to avoid hypoglycemia, and an approach to correction of hyper- and hypoglycemia during the preparation (Appendix A). Patients were prescribed a polyethylene glycol (PEG)-based osmotic laxative (PEG-3350, Sodium Sulfate, Sodium Chloride, Potassium Chloride, Sodium Ascorbate, and Ascorbic Acid for Oral Solution) for bowel cleaning. 

### 2.6. Statistical Analysis

Descriptive statistics were performed for all study variables. For group variables, we presented numbers and percentages from the whole study sample. We used a one-sample Kolmogorov-Smirnov test and the graphical presentation of data by the histogram to check the normality of this study variables. As all the continuous variables were distributed differently than normal, we presented medians and interquartile ranges (IQR). 

We performed the univariate analysis to assess the difference in socio-demographic variables between this study groups (we used the Chi-square test for the group variables and the Mann-Whitney test for continuous variables). 

We calculated the mean value from the 17 questions of the gastrointestinal symptom score to assess the mean frequency and intensity of all investigated gastrointestinal symptoms, as well as the mean value of the frequency of use of medications. The Cronbach Alpha test was applied to check the reliability of this scale. 

In addition, we evaluated the answers to questions about upper and lower gastrointestinal disease, history of liver and pancreatic diseases, history of gastrointestinal malignancy, and history of abdominal surgery. 

We investigated the difference between those with stable and progressive DKD in gastrointestinal symptoms (Cramer’s V test), symptom intensity (Mann-Whitney test), use of medications (Mann-Whitney test), and presence and history of diseases (Chi-square test). 

A Spearman correlation was performed to identify the relationship between markers of kidney disease and demographic and blood/urine parameters. The *p* values < 0.05 were considered statistically significant. 

We built multiple logistic regression models to assess the association between the progression of DKD and gastrointestinal symptoms. We adjusted the model for the variables that displayed statistical significance in the univariate analysis or had clinical relevance. The full adjustment set included sex, diabetes duration, HbA1c, and BMI.

## 3. Results

### 3.1. Description of this Study Groups and Inflammatory Markers

This study sample consists of 100 patients with T1D, mostly women (62%) with a mean age of 42.59 ± 13.18 and a mean BMI of 25.98 ± 4.69 1 kg/m^2^. The mean follow-up time in the cohort is 5.89 ± 1.73 years. Most of the participants were non-smokers (63%), and they had hypertension (53%). The mean duration of diabetes was 24.38 ± 11.92 years, and the mean HbA1c was 8.28 ± 1.74%. 52 (52%) of subjects had retinopathy, 27 (27%) had autoimmune thyroid disease, and 16 (16%) had CVD. Study groups (with stable DKD disease and with progressive DKD) differed by smoking (more smokers among those with progressive DKD), hypertension (more patients with hypertension among those with progressive DKD), length of diabetes (longer history of diabetes for patients with progressive DKD), retinopathy, HbA1c, CVD, eGFR, albuminuria, serum bilirubin, blood haemoglobin concentration, and erythrocyte counts. The level of calprotectin in faeces and serum CRP did not differ between the groups of patients with progressive DKD and patients with stable DKD statistically significantly (Table 1). Moreover, of the 100 patients in this study, only 10 had faecal calprotectin ≥50 µg/g.

### 3.2. Gastrointestinal Diseases and Symptoms

The gastrointestinal symptom score demonstrated high reliability (Cronbach’s α = 0.78). Patients with progressive DKD had higher gastro-intestinal symptoms scores (Appendix A) compared to those with stable DKD (*p* = 0.019). In addition, 14 (52%) of patients with progressive DKD had bowel movement disorders, versus 16 (22%) in patients with stable DKD (*p* < 0.01) (Table 2.)

There was no statistically significant difference between the patients with progressive DKD and patients with stable DKD on previous upper and lower gastrointestinal disease (*p* = 0.50 and *p* = 0.35, respectively). There was no difference between groups in the history of liver and pancreatic diseases (*p* = 0.74) or in the history of gastrointestinal malignancy (*p* = 0.10). The history of abdominal surgery was more frequent in patients with progressive DKD at the significance level of *p* = 0.07. There were no statistically significant differences between DKD patients with progressive DKD and patients with stable DKD in their use of medications for gastrointestinal disorders (*p* = 0.54). 

### 3.3. Correlation between Gastrointestinal Symptom Scores with Clinical Markers and Regression Analysis

We investigated correlations between the gastrointestinal symptom score and several clinical markers in this study sample. The scores correlated negatively with eGFR (r = −0.335; *p* = 0.001), weight (r = −0.236, *p* = 0.018), blood erythrocyte counts (r = −0.313, *p* = 0.002), and blood haemoglobin (r = −0.321, *p* = 0.001) and positively with albuminuria (r = 0.245; *p* = 0.015), Hba1c (r = 0.305, *p* = 0.002), and diabetes duration (r = 0.251, *p* = 0.012).

Faecal calprotectin did not correlate with the gastrointestinal symptom score, eGFR, or albuminuria.

In univariate regression analysis, higher scores in gastrointestinal symptoms were associated with higher odds of DKD progression (3.086 (1.209, 7.879), *p* = 0.018). The association remained significant when the model was adjusted for sex and BMI. However, the association was no longer significant when the model was adjusted for sex, BMI, diabetes duration, and HbA1c, with diabetes duration remaining the only significant predictor of DKD (odds ratio 1.058 (1.011, 1.106), *p* = 0.014) in the model and HbA1c demonstrating association at a significance level of 0.075 (odds ratio 1.302 (0.978, 1.734). The results of the logistic regression analysis are summarised in Table 3.

### 3.4. Endoscopy

Prevalence of indications for colonoscopy did not differ between the patients with and without DKD progression.

From the 47 patients who were selected to undergo colonoscopy, 13 had progressive DKD and 34 did not (48.1% and 46.5% of the progressive DKD and non-progressive DKD groups, respectively). Only 21 patients (4 patients with progressive DKD and 17 patients with stable DKD) accepted the invitation and went through the procedure. The reasons for non-acceptance of the invitation included a complicated preparation procedure for the colonoscopy with the necessity of fasting and intensive glucose control, a fear of hypoglycemia, and a fear of complications during anaesthesia. Many patients refused to undergo the investigation because of their poor health status due to complications of diabetes and co-morbidities. 

Among patients who underwent the endoscopic examination, the most frequent indications for endoscopic examination were abdominal pain (n = 17; 81%), bowel movement disorders (n = 9; 43%), and elevated faecal calprotectin (>50 µg/g) (n = 5; 24%).

Of the 10 individuals who underwent upper GI endoscopy, 7 (70%) had abnormal macroscopic findings and were analysed for H. pylori infection by rapid urease tests (RUT). None of the patients had a positive RUT; however, histopathology identified two patients with *H. pylori*, visible on special staining. Most of the gastric lesions were minor endoscopic findings (Table 4). Of the five endoscopically reported hyperaemic gastropathy and duodenopathy patients, two had active gastritis and two had chronic atrophic gastritis. Of the three patients with macroscopically normal upper GI endoscopy, one had chronic atrophic gastritis and one had active gastritis with erosive gastropathy. 

In total, 21 colonoscopies were performed. Five (24%) of the colonoscopies performed showed abnormal macroscopic findings (Table 4, Appendix A). The most commonly reported histopathological findings were infiltration with eosinophils, lymphocytes, plasmacytes, the presence of lymphoid follicles, and lymphoid aggregates. We also observed stromal fibrosis, mononuclear cell and macrophage infiltration, and tubular adenomas in several patients. Active inflammation was found in six patients, and one patient had active colitis on histopathological examination. No malignancies were found. 

## 4. Discussion

The main finding of our study is that in patients with T1D and progressive DKD, the frequency and severity of gastrointestinal symptoms are higher compared to patients with stable kidney markers, as assessed by the gastrointestinal symptom score. The link between progressive DKD and diabetic gastroenteropathy was additionally confirmed by statistically significant correlations between gastrointestinal symptom scores, eGFR, and albuminuria, as well as in the logistic regression models. 

We demonstrate in our work that gastrointestinal disorders start to manifest already in the initial stages of DKD progression, as the median eGFR is 72.81(40.72–105.27) mL/min/1.73 m^2^ in the group of progressive DKD in our study. Our findings cannot be directly compared to previous data due to the deficiency of studies on the association of diabetic gastroenteropathy with progressive DKD in T1D. We only found one paper by D’Addio and colleagues reporting higher scores of gastrointestinal symptoms in patients with T1D and DKD. However, that paper investigated diabetic gastroenteropathy only in patients with ESRD compared to healthy individuals and did not cover subjects with DKD or the initial stages of chronic kidney disease [12]. More pronounced gastrointestinal symptoms in ESRD from any cause were already reported and might result from metabolic disorders and treatment of ESRD [27]. D’Addio and colleagues also report that the gastrointestinal symptoms of patients subsided significantly several years after successful treatment with kidney-pancreas transplantation. However, this was not the case in patients who only received kidney transplants, indicating hyperglycemia per se as the main factor in diabetic gastroenteropathy. Although we think that immunosuppressive treatment after transplantation might be one of the factors alleviating the gastrointestinal derangements in the above study, our results of the regression analysis with a fully adjusted model agree with the latter finding. Specifically, after adjustment of the regression model for diabetes duration, gastrointestinal symptom scores were no longer significant predictors of progressive DKD. We suggest that a larger longitudinal study is needed to obtain more conclusive results about mutual associations between gastrointestinal derangements and DKD progression in T1D. Such a study is necessary for the development of future approaches to DKD prevention and treatment. Indeed, gastrointestinal symptoms are usually a sign of dysfunction in the gastrointestinal tract. Therefore, they might be associated with impaired nutritional balance and increased gut permeability, resulting in augmented low-grade inflammation, which has been shown to be associated with DKD progression [7,8,9,10,33,34].

In contrast to some previous studies in T1D reporting higher calprotectin levels in patients with DKD [10,11,35], we did not observe higher faecal calprotectin levels in patients with progressive DKD compared to subjects with stable kidney markers. Moreover, faecal calprotectin did not exhibit any correlations with gastrointestinal symptom scores, eGFR, or albuminuria in our study. Our findings might mean that gastrointestinal symptoms in our patients were not associated with neutrophil-mediated inflammation, resulting in an increase in calprotectin and a severe inflammatory reaction, which is observed in inflammatory bowel disease [36]. Indeed, in the histological investigation of the colon biopsies of patients who underwent endoscopic examination, eosinophilic infiltration, lymphoid follicles/lymphoid aggregates, and lymphoplasmacytic infiltration were the most frequent findings. In 21 subjects who underwent colonoscopy, infiltration with polymorphonuclear neutrophils was not registered, mononuclear cell infiltration was observed only in two patients, and active inflammation was reported only in six subjects (only one of them with progressive DKD). Thus, histologic changes observed by us indicate increased reactivity of the mucosa and chronic pathology [37], when increased calprotectin levels are unlikely. On the other hand, lymphoplasmacytic infiltration is a predictor of more severe colitis in the future [37], and a study in T1D with ESRD reports more severe histologic changes of the mucosa as compared to our findings [12]. Thus, it is possible that an increase in calprotectin is observed in more advanced DKD stages.

Our study’s limitations include a relatively low number of subjects, especially in the group of progressive DKD, which might have influenced the results. Further, the cross-sectional nature of this study does not allow us to evaluate the causal relationship between symptoms of diabetic gastroenteropathy and DKD. Other limitations include the self-evaluation of gastrointestinal symptoms in this study and the self-reporting of previous gastrointestinal diseases in a subset of patients due to unavailable central medical records in Latvia. We also did not perform an evaluation of autonomic neuropathy, which might have influenced the gastrointestinal symptoms; however, to our opinion, these data would not alter our conclusions or the clinical value of our study due to the availability of treatment for autonomic neuropathy nowadays. Lack of endoscopic evaluation in all study participants is another limitation. However, all study participants did not have indications for endoscopy. In addition, not all of the invited patients responded to the invitation due to complicated preparation procedures, fear of hypoglycemia, fear of complications during anaesthesia, or generally poor health status. 

The major strength of this study is the analysis of gastrointestinal symptoms in patients with different rates of progression of DKD, in contrast to previous studies addressing either patients with T1D generally or patients with T1D and ESRD in comparison to healthy subjects. In addition, we have demonstrated results on histological colon biopsies in 21 patients with T1D, previously described only in one study [12].

## 5. Conclusions

To conclude, in patients with T1D and progressive DKD, the frequency and severity of gastrointestinal symptoms are higher compared to patients with stable kidney markers, as assessed by gastrointestinal symptom scores. Moreover, gastrointestinal symptom scores correlate with kidney markers. Further research is needed to clarify the causal relationships of the gut-kidney axis in T1D. 

## Figures and Tables

**Figure 1 biomedicines-11-02679-f001:**
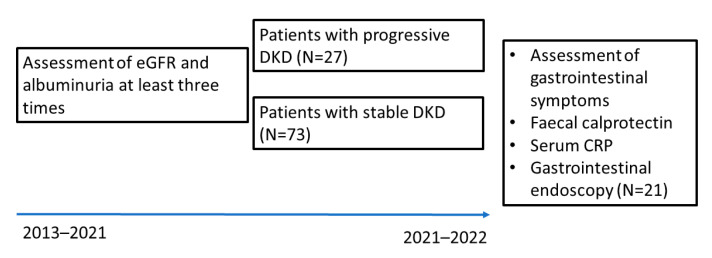
Recruitment scheme. eGFR—estimated glomerular filtration rate; DKD—diabetic kidney disease; CRP—C-reactive protein.

**Table 1 biomedicines-11-02679-t001:** Differences in socio-demographic and disease-related data among study groups.

	DKD Stable, N = 73	DKD Progressive, N = 27	*p* Value
Male gender, N (%)	31 (42.5%)	7 (25.9%)	0.99
Age, years,	44.0 (31.5–52.0)	39.0 (34.0–50.0)	0.86
BMI, kg/m^2^	24.85 (22.63–28.15)	25.40 (22.60–28.40)	0.95
Smokers, N (%)	13 (17.8%)	6 (22.2%)	<0.01
Hypertension, N (%)	32 (43.8%)	21 (77.8%)	<0.01
Length of diabetes, years	21 (13–31)	27 (24–37)	<0.01
Mean follow-up, years	7.00 (5.50–7.00)	7.00 (3.00–7.00)	0.72
Retinopathy, N (%)	30 (41.1%)	22 (81.5%)	<0.01
Cardiovascular disease, N (%)	7 (9.6%)	9 (33.3%)	<0.01
On ACEI/ARB, N (%)	12 (16.4%)	8 (29.6%)	0.14
On lipid lowering medication, N (%)	17 (23.3%)	10 (37.0%)	0.16
Autoimmune thyroid disease, N (%)	18 (24.7%)	9 (33.3%)	0.27
Other autoimmune disease, N (%)	11 (15.1%)	5 (18.5%)	0.67
Haemoglobin A1C, %	7.60 (7.00–8.70)	8.90 (07.54–10.30)	0.02
Haemoglobin A1C, mmol/mol	59.56 (53.00–71.58)	73.77 (58.90–89.07)	0.02
Estimated glomerular filtration rate, mL/min/1.73 m^2^	107.98(92.52–117.63)	72.81(40.72–105.27)	<0.01
End stage renal disease, N (%)	0 (0%)	4 (14.8%)	<0.01
Albumin/creatinine ratio in urine, mg/mmol	0.45(0.19–0.99)	19.65 (4.98–112.50)	<0.01
C-reactive protein, mg/L	0.9 (0.5–2.8)	1.5 (0.5–3.48)	0.72
Faecal calprotectin, µg/g	7.60 (2.75–23.20)	16.20 (5.80–22.00)	0.21
Faecal calprotectin > 50 µg/g, N (%)	8 (11.0%)	2 (7.4%)	0.59
Total cholesterol, mmol/L	5.02(4.30–5.79)	4.96(4.12–5.88)	0.95
Low density lipoproteins, mmol/L	2.87(2.10–3.40)	3.04(2.10–3.46)	0.41
Triglycerides, mmol/L	1.14(0.85–1.45)	1.31(0.86–2.11)	0.14
Alanine transaminase, U/L	19.00(15.50–29.00)	21.00(17.00–24.00)	0.64
Gamma-glutamyl Transferase, U/L	16.00(13.00–25.50)	17.00(15.00–26.00)	0.41
Bilirubin, µmol/L	9.30(7.80–12.40)	6.51(5.10–9.20)	0.001
Haemoglobin, g/L	140.00(132.00–150.00)	129.00(120.00–141.00)	0.001
Erythrocytes, 10 × 12/L	4.70(4.40–5.00)	4.40(4.00–4.73)	0.002
Leukocytes, 10 × 9/L	6.13(5.01–7.22)	6.53(5.71–7.64)	0.14
Thrombocytes, 10 × 9/L	259.00(228.00–284.00)	231.00(200.00–299.00)	0.21

Continuous variables are presented as medians (IQR). DKD—diabetic kidney disease. eGFR—estimated glomerular filtration rate. Diabetic retinopathy—history of any stage of retinopathy based on medical recordings. Arterial hypertension—systolic blood pressure ≥ 140 mmHg (18.7 kPa) or diastolic blood pressure ≥ 90 mmHg (12.0 kPa), or a history of antihypertensive drug treatment. Autoimmune thyroid disease—Hashimoto’s thyroiditis or Graves’ disease Other autoimmune diseases—history of autoimmune rheumatologic diseases such as rheumatoid arthritis, sacroiliitis, psoriasis, asthma, etc. CVD—cardiovascular disease, defined as a history of acute myocardial infarction, coronary bypass/percutaneous transluminal coronary angioplasty stroke, amputation, or peripheral vascular disease.

**Table 2 biomedicines-11-02679-t002:** Gastrointestinal conditions and symptoms in this study groups.

	DKD Stable, N = 73	DKD Progressive, N = 27	*p* Value
History of upper gastrointestinal disease, N (%)	29 (39.7%)	13 (48.1%)	0.50
History of lower gastrointestinal disease, N (%)	24 (32.9%)	12 (44.4%)	0.28
History of gastrointestinal malignancy, N (%)	0 (0.0%)	1 (3.7%)	0.10
History of liver and pancreas disease, N (%)	27 (37.0%)	9 (33.3%)	0.74
History of abdominal surgery, N (%)	21 (28.8%)	13 (48.1%)	0.07
Mean value of gastrointestinal symptom score	1.05 (0.73–1.35)	1.27 (1.00–1.82)	0.019
Bowel movement disorders, N (%)	16 (21.9%)	14 (51.9%)	<0.01
Usage of medications for gastrointestinal disorders	0.16 (0.00–0.50)	0.33 (0.00–0.66)	0.54

Continuous variables are presented as medians (IQR). Bowel movement disorders—constipation, diarrhoea, intermittent constipation, and diarrhoea. DKD—diabetic kidney disease.

**Table 3 biomedicines-11-02679-t003:** Association between the progression of diabetic kidney disease and gastrointestinal symptoms.

Variable	Model	Odds Ratio, OR	95% Confidence Interval (CI)	*p* Value
Mean Symptoms	1	3.086	1.20; 7.87	0.02
Mean Symptoms	2	2.77	1.06; 7.24	0.04
Mean symptoms	3	2.394	0.886; 6.467	0.085
Mean Symptoms	4	1.99	0.71; 5.54	0.18

Results of the logistic regression analysis with the presence of progressive diabetic kidney disease as the response variable. Data are presented as odds ratios with 95% CI and *p*-values. Model 1—univariate. Model 2 was adjusted for sex and BMI. Model 3 was adjusted for sex, BMI, and diabetes duration. Model 4 was adjusted for sex, BMI, diabetes duration, and HbA1c.

**Table 4 biomedicines-11-02679-t004:** Summary of macroscopic and microscopic lesions identified during gastrointestinal endoscopic examination.

Procedure	Macroscopic Lesions	Histopathologic Lesions
Upper endoscopy, N = 10	(1) Hyperaemic gastropathy (n = 4) (2) Hyperaemic duodenopathy (n = 1) (3) oesophageal candidiasis (n = 1) (4) gastric intestinal metaplasia (n = 1)	(1) active gastritis (N = 3) (2) chronic atrophic gastritis (N = 3)
Colonoscopy, N = 21	(1) polyps (n = 2) (2) diverticulosis (n = 1) (3) erosion of sigmoid colon (n = 1) (4) perianal papilloma (n = 1)	(1) eosinophilic infiltration (n = 8, 38%) (2) lymphoid follicles/lymphoid aggregates (n = 7, 33%) (3) lymphoplasmacytic infiltration (n = 5, 24%). (4) lymphocyte and macrophage infiltration (n = 3, 14%) (5) stromal fibrosis (n = 3, 14%) (6) mononuclear cell infiltration (n = 2, 9.5%) (7) tubular adenomas (n = 2, 9.5%). (8) Active inflammation 6 (29%) (9) active colitis 1 (4.76%)

## Data Availability

All data are not available publicly due to privacy restrictions. The data are available from the corresponding author on request.

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
