# Peer review of "Progression of Diabetic Kidney Disease and Gastrointestinal Symptoms in Patients with Type I Diabetes"

_biomedicines, 2023, doi:10.3390/biomedicines11102679_

Round 1

Reviewer 1 Report

In this article "Progression of diabetic kidney disease and gastrointestinal symptoms in patients with type I diabetes", Aleksejs et al., focused on diabetic kidney disease (DKD) and gastrointestinal symptoms, and aim to link gastrointestinal symptoms with DKD progression. All the results provide further insight into analyse the prevalence of gastrointestinal symptoms, history of previous gastrointestinal conditions, data of endoscopic gastrointestinal investigations and levels of faecal calprotectin in patients with T1D and different velocity of DKD progression in a cross-sectional study. Overall, this article is well organized and designed. And the conclusions sound convincing. It is a good job.

However some minor revisions are needed. 

In Abstract part, some spellings needs corrections.

In keywords part, DKD needs to be added.

In some tables, "Title 1" is needless. 

Author Response

Response to Reviewer 1 Comments

Thank you very much for taking the time to review this manuscript. Please find the detailed responses below and the corresponding revisions/corrections in track changes in the re-submitted files

1). In Abstract part, some spellings needs corrections.

We carefully checked the spelling and corrected where necessary

2) In keywords part, DKD needs to be added.

We added diabetic kidney disease (DKD) to keywords

3) In some tables, "Title 1" is needless. 

Thank you for noticing, we removed “Title 1” from the first column of Table 1 and Table 2

Reviewer 2 Report

I noticed only few issues to revise in this quality manuscript. 

- Abstract, introduction and methods are adequate. So the statistics except lack of statement of normality analysis. Since all variables were expressed as medians I assume that none of the variables were fit into normal distribution. Yet, I suggest authors to express the normality test that conducted in statistical analyses.

- Gastrointestinal endoscopy results could  be expressed separately for both groups, not whole study population

- Discussion can be improved. Inflammation is a characteristic feature of diabetic kidney disease (Postgraduate Medicine 2023;135(5):519-523. DOI:10.1080/00325481.2023.2214058). Gastrointestinal inflammation is detected in some of the participants which may cause nutritional defects. Moreover, nutritional markers, i.e. prognostic nutritional index, is associated with diabetic kidney disease (J. Clin. Med. 2023, 12, 5952. https://doi.org/10.3390/jcm12185952). All of these markers are considered as inflammatory predictors, therefore, authors should emphasize the role of inflammation on study results. 

Author Response

Response to Reviewer 2 Comments

Thank you very much for taking the time to review this manuscript. Please find the detailed responses below and the corresponding revisions/corrections in track changes in the re-submitted files.

1). So the statistics except lack of statement of normality analysis. Since all variables were expressed as medians I assume that none of the variables were fit into normal distribution. Yet, I suggest authors to express the normality test that conducted in statistical analyses.

We apologize for not adding information on normality testing into the manuscript. We used one-sample Kolmogorov-Smirnov test for normality analysis. As sometimes this test gives a very strong result, and the normality can be assumed under some assumptions even in case the Kolmogorov-Smirnov test is significant, we used additionally a graphical presentation (histogram) of the variables included in the analysis for the check of normality. This information is now included in Section 2.6. “Statistical analysis”. The added text is (underlined):

“We used a one-sample Kolmogorov-Smirnov test and the graphical presentation of data by the histogram to check the normality of the study variables. As all the continuous variables were distributed other than normal, we presented medians and interquartile ranges (IQR).”

2). Gastrointestinal endoscopy results could be expressed separately for both groups, not whole study population

Thank you for this suggestion. We added such a Table to the supplements (Supplement 3). We would like to note that the number of subjects with progressive DKD who underwent colonoscopy was only 4, which makes the comparisons between groups by statistical methods impossible.

3) Discussion can be improved. Inflammation is a characteristic feature of diabetic kidney disease (Postgraduate Medicine 2023;135(5):519-523. DOI:10.1080/00325481.2023.2214058). Gastrointestinal inflammation is detected in some of the participants which may cause nutritional defects. Moreover, nutritional markers, i.e. prognostic nutritional index, is associated with diabetic kidney disease (J. Clin. Med. 2023, 12, 5952. https://doi.org/10.3390/jcm12185952). All of these markers are considered as inflammatory predictors, therefore, authors should emphasize the role of inflammation on study results. 

Thank you very much for these interesting and useful suggestions. We want to note that we touched upon low grade inflammation in association with DKD in the introduction, as a rationale of our study. We added to the discussion information on inflammation as a potential consequence of gastrointestinal derangements in T1D, using the references you kindly suggested. However, we would like to refrain from adding more information on low grade inflammation. Serum inflammatory markers were not studied in our work and thus broad information about this topic might confuse the reader about the focus of our study.

Added to the discussion: “We suggest that a larger longitudinal study is needed to obtain more conclusive results about mutual associations between gastrointestinal derangements and DKD progression in T1D. Such study is necessary for development of future approaches for DKD prevention and treatment. Indeed, gastrointestinal symptoms usually are a sign of dysfunction of the gastrointestinal tract. Therefore, they might be associated with impaired nutritional balance ( Acta Diabetol 2023 Feb;60(2):235-245. doi: 10.1007/s00592-022-01985-x. Epub 2022 Nov 2. ) and increased gut permeability resulting in augmented low grade inflammation (Postgrad Med. 2023 Jun;135(5):519-523.  doi: 10.1080/00325481.2023.2214058. Epub 2023 May 16. ), which have been shown to be associated with DKD progression “